# FGF14 modulates resurgent sodium current in mouse cerebellar Purkinje neurons

**Haidun Yan[1,2], Juan L Pablo[2,3], Chaojian Wang[1,2], Geoffrey S Pitt[1,2,3]***

[1]Department of Medicine, Duke University Medical Center, Durham, United States; [2]Ion Channel Research Unit, Duke University Medical Center, Durham, United States; [3]Department of Neurobiology, Duke University Medical Center, Durham, United States

**Abstract** Rapid firing of cerebellar Purkinje neurons is facilitated in part by a voltage-gated Na$^+$ (Na$_V$) 'resurgent' current, which allows renewed Na$^+$ influx during membrane repolarization. Resurgent current results from unbinding of a blocking particle that competes with normal channel inactivation. The underlying molecular components contributing to resurgent current have not been fully identified. In this study, we show that the Na$_V$ channel auxiliary subunit FGF14 'b' isoform, a locus for inherited spinocerebellar ataxias, controls resurgent current and repetitive firing in Purkinje neurons. FGF14 knockdown biased Na$_V$ channels towards the inactivated state by decreasing channel availability, diminishing the 'late' Na$_V$ current, and accelerating channel inactivation rate, thereby reducing resurgent current and repetitive spiking. Critical for these effects was both the alternatively spliced FGF14b N-terminus and direct interaction between FGF14b and the Na$_V$ C-terminus. Together, these data suggest that the FGF14b N-terminus is a potent regulator of resurgent Na$_V$ current in cerebellar Purkinje neurons.

*For correspondence: geoffrey.pitt@duke.edu

**Competing interests:** The authors declare that no competing interests exist.

**Reviewing editor**: Gary L Westbrook, Vollum Institute, United States

## Introduction

Cerebellar Purkinje neurons, which provide the sole output from the cerebellar cortex, display a repetitive firing behavior driven by voltage-gated Na$^+$ channels (Na$_V$ channels). Fast repetitive firing in Purkinje neurons is promoted by unusual characteristics associated with Na$_V$ channel inactivation, whereby channels recover unusually rapidly from inactivation and in doing so pass a 'resurgent' sodium current, which helps drive the cell to fire a subsequent action potential. The molecular components underlying the peculiar properties of Na$_V$ channels in Purkinje neurons have not been definitively identified.

Several features of Na$_V$ currents in cerebellar Purkinje neurons, however, set them apart from Na$_V$ currents recorded in other neurons. First, more than half of the Na$_V$ current in Purkinje neurons is carried by Na$_V$1.6, an isoform for which inactivation is less complete compared to other Na$_V$ channels (*Raman et al., 1997*). Examination of Na$_V$ currents in cerebellar Purkinje neurons from *Scn8a$^{-/-}$* mice showed that, compared to the other resident Na$_V$ channels, the *Scn8a*-encoded Na$_V$1.6 channels have a relatively large 'late' (non-inactivating) component and Na$_V$1.6 channels have increased availability (depolarized V$_{1/2}$ for steady-state inactivation). Second, Na$_V$ channels in cerebellar Purkinje neurons display an unusual transient re-opening during repolarization, which is identified as 'resurgent' Na$_V$ current (*Raman and Bean, 1997*). This resurgent current derives from an unblocking of open channels by a peptide moiety that can be eliminated by the intracellular application of trypsin or chymotrypsin proteases into the cytoplasm (*Grieco et al., 2005*). The blocking particle acts only on open channels and competes with the inactivation process to prevent channels from entering an absorbed, inactivated state. During action potential repolarization, unblocking of these channels then allows Na$^+$ influx that can initiate a subsequent action potential (*Khaliq et al., 2003*). Since the Na$_V$ current in Purkinje

**eLife digest** The cerebellum is a region of the brain that is involved in motor control, and it contains a special type of nerve cells called Purkinje neurons. Messages travel along neurons as electrical signals carried by sodium ions, which have a positive electric charge. Normally, when a neuron is 'at rest', the plasma membrane that surrounds the neuron prevents the sodium ions outside the cell from entering. To send an electrical signal, voltage-sensitive proteins in the membrane called sodium channels open up. This allows the sodium ions to enter the cell by passing through a pore in the channel protein, thereby changing the voltage across the membrane.

Once sodium channels open, they rapidly become 'locked' in a closed state, which allows the membrane voltage to return to its original value before another signal can be sent. This locked state also prevents sodium channels from reopening quickly. As a consequence most neurons cannot send successive electrical signals rapidly. Purkinje neurons are unusual because they can send many electrical signals in quick succession—known as rapid firing—without having to be reset each time.

Rapid firing is possible in Purkinje neurons because the channel proteins can be reopened to allow more $Na^+$ to enter the cell, but it is not clear how this is controlled. Now, based on experiments on Purkinje neurons isolated from mice, Yan et al. have shown that a protein called FGF14 that binds to the sodium channel proteins can help them to reopen quickly in order to allow rapid firing.

Spinocerebellar ataxia is a degenerative disease caused by damage to the cerebellum that leads to loss of physical coordination. Some patients suffering from this disease carry mutations in the gene that makes the FGF14 protein. Therefore, understanding the role of FGF14 in the rapid firing of Purkinje neurons may aid the development of new treatments for this disease.

neurons carried by $Na_V1.6$ is relatively resistant to inactivation, it is particularly susceptible to open block by the peptide moiety. Why $Na_V1.6$ channels in cerebellar Purkinje neurons are relatively resistant to inactivation, however, is unknown.

Here, we focused on the role of the ion channel regulator FGF14, which is enriched in cerebellar granule and Purkinje neurons (*Wang et al., 2002*). FGF14 is one of four fibroblast growth factor homologous factors (FHFs), members of the fibroblast growth factor (FGF) superfamily sharing a FGF-like core but having extended N- and C-termini not found in other FGFs. Individual FHFs undergo alternative splicing that generates distinct N-termini, none of which contain a signal sequence (*Smallwood et al., 1996*; *Munoz-Sanjuan et al., 2000*). Thus, unlike other FGFs they are not secreted and do not function as growth factors (*Smallwood et al., 1996*). Instead, FHFs remain intracellular and modulate various ion channels. FHFs can bind directly to the cytoplasmic C-terminal domains (CTDs) of $Na_V$ channels (*Liu et al., 2001*) and regulate $Na_V$ channel function (*Lou et al., 2005*). Further, FGF14 is a potent regulator of both $Na_V1.1$ and $Na_V1.6$ (*Lou et al., 2005*; *Laezza et al., 2009*), the two dominant $Na_V$ channels in cerebellar Purkinje neurons (*Raman et al., 1997*; *Kalume et al., 2007*). FHFs also regulate voltage-gated $Ca^{2+}$ channels through a mechanism that does not appear to involve a direct interaction (*Hennessey et al., 2013*; *Yan et al., 2013*). An ataxia phenotype in *Fgf14*$^{-/-}$ mice highlights the role of FGF14 in cerebellar physiology and the development of spinocerebellar ataxia type 27 (SCA27) in patients with either FGF14 haploinsufficiency or a dominant negative *FGF14* mutation (*Wang et al., 2002*; *van Swieten et al., 2003*; *Dalski et al., 2005*) underscores the role of FGF14 in disease.

Using shRNA knockdown of endogenous FGF14 in cultured cerebellar Purkinje neurons and replacement with informative mutants that blocked binding of FGF14 to $Na_V$ CTDs and eliminated the FGF14 extended N-terminus, we found that FGF14, and especially the N-terminus of the FGF14b isoform exerts specific kinetic effects on cerebellar $Na_V$ channels that support repetitive firing and therefore provides new information on the pathophysiology of SCA27.

## Results

To test whether FGF14 was a contributor to regulation of $Na_V$ current in cerebellar Purkinje neurons, we first confirmed its expression in cerebellum and demonstrated that FGF14 co-immunoprecipitates with $Na_V1.6$ channels in mouse cerebellar lysates (*Figure 1A*). Although FGF14 was previously shown

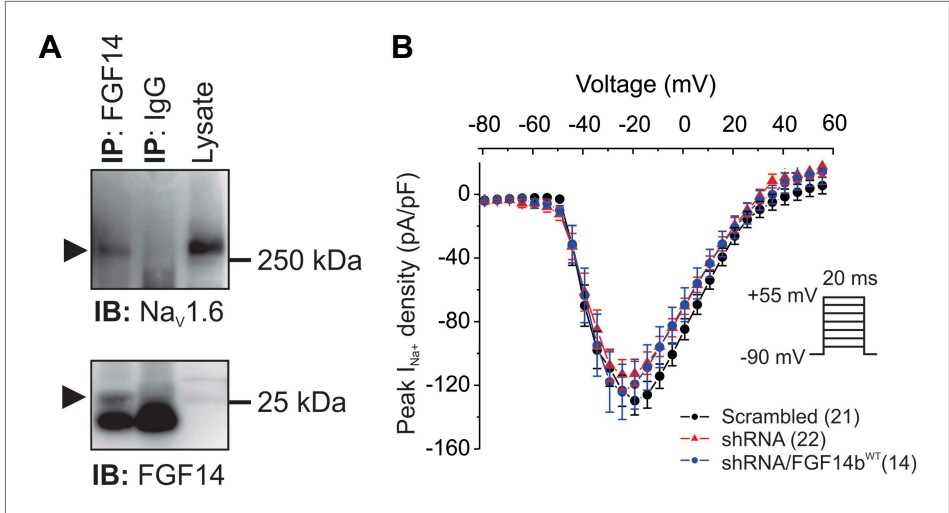

**Figure 1**. FGF14 is a component of the Na$_V$1.6 macromolecular complex in mouse cerebellum, but shRNA knockdown does not affect Na$_V$ current density in cultured cerebellar Purkinje neurons. (**A**) Co-immunoprecipitation (IP) of Na$_V$1.6 by FGF14 from mouse cerebellum and immunoblot (IB) with the indicated antibodies. M$_w$ markers are indicated on right. Arrowheads indicate Na$_V$1.6 (top panel) and FGF14 (bottom panel). Intense signal in bottom panel below the FGF14 signal is immunoglobulin light chain. (**B**) Current–voltage relationships (normalized to cell capacitance) of transient Na$_V$ currents from cerebellar Purkinje neurons transfected with Scrambled control shRNA (Scrambled), FGF14 shRNA (shRNA), or FGF14 shRNA plus the shRNA-resistant FGF14b$^{WT}$ (shRNA/FGF14$^{WT}$). The number of neurons tested, *N*, is in parentheses. Inset shows schematic of voltage protocol.

to co-immunoprecipitate with Na$_V$1.6 in a heterologous expression system (*Laezza et al., 2009*), co-immunoprecipitation between FGF14 and Na$_V$1.6 channels has not been reported from brain.

To evaluate the consequences of FGF14 haploinsufficiency in SCA27, we used a previously validated shRNA (*Yan et al., 2013*) to knockdown endogenous FGF14 in cultured cerebellar Purkinje neurons. Knockdown of FGF14 did not affect Na$_V$ current density compared to the Na$_V$ current density recorded in neurons transfected with a control shRNA (*Figure 1B*). Because knockout of *Fgf14* reduced spiking activity (*Goldfarb et al., 2007*) but Na$_V$ current density was unaffected in *Fgf14* knockout or knockdown, we hypothesized that the absence of FGF14 must affect Na$_V$ kinetic properties in ways that prevent repetitive firing. We therefore examined Na$_V$ kinetic properties in more detail.

Voltage dependence of activation was unaffected by FGF14 knockdown (*Figure 2A*), but the V$_{1/2}$ of steady-state inactivation was shifted about −10 mV, thereby decreasing channel availability (*Figure 2B* and *Table 1*). When we co-expressed a FGF14b cDNA with synonymous substitutions rendering it resistant to the FGF14 shRNA (FGF14b$^{WT}$), the V$_{1/2}$ of steady-state inactivation was restored to the control voltage; activation was unaffected (*Figure 2A–B* and *Table 1*). The 'rescue' of steady-state inactivation by expression of the shRNA-insensitive FGF14 provided additional demonstration of the specificity of our shRNA targeting strategy (and see similar rescue for additional parameters below). In addition to the decreased channel availability, we observed that FGF14 knockdown accelerated inactivation kinetics (*Figure 2C–D*). FGF14 knockdown also reduced the late Na$_V$ current (*Figure 2E–F*). Thus, by several different kinetic measures, FGF14 knockdown in cerebellar Purkinje neurons biased the endogenous Na$_V$ channels towards entering the inactivated state. We anticipated, therefore, that FGF14 knockdown should reduce Na$_V$ resurgent current amplitude, a key feature that allows cerebellar Purkinje neurons to fire repetitively. Indeed, the amplitude of the resurgent Na$_V$ current (*Figure 2G–J*) was markedly diminished using a well-established protocol to elicit Na$_V$ resurgent current (*Raman and Bean, 1997*). As with the other kinetic properties of Na$_V$ currents (above), co-expression of FGF14b$^{WT}$ in the presence of the shRNA provided a complete rescue (*Figure 2G–J*). While FGF14 knockdown reduced the amplitude of resurgent Na$_V$ current, it did not significantly affect the time to peak (*Figure 2K*).

To determine how FGF14 regulates Na$_V$ kinetics and Na$_V$ resurgent current, we turned our focus to the FGF14 N-terminus. The alternatively spliced N-termini of various FHFs have been shown to exert

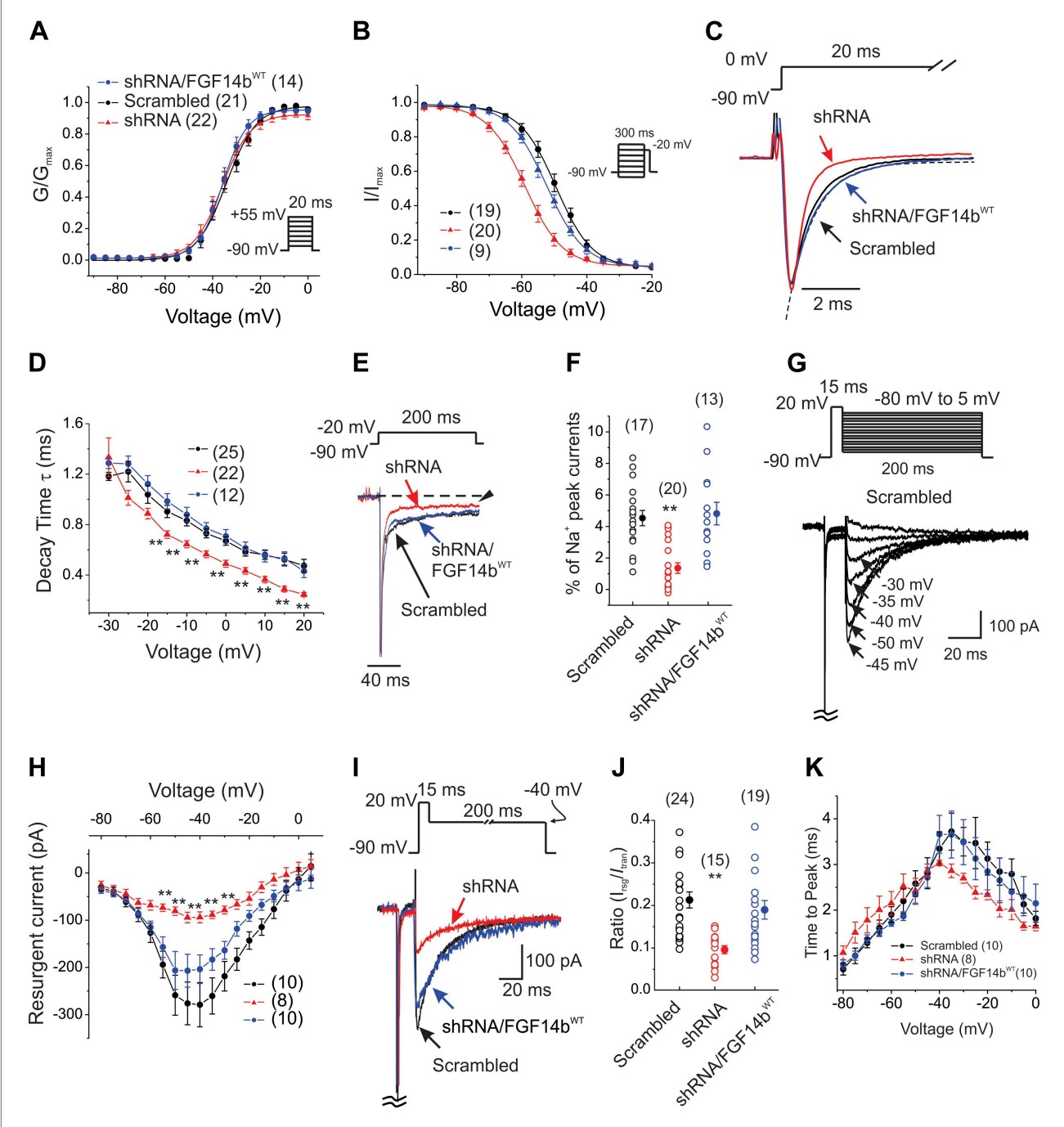

**Figure 2**. FGF14 knockdown in cerebellar Purkinje neurons affects multiple Na$_V$ channel biophysical properties. (**A** and **B**) Voltage-dependence of Na$_V$ channel activation and steady-state inactivation in cerebellar Purkinje neurons transfected with Scrambled control shRNA (Scrambled), FGF14 shRNA (shRNA), or FGF14 shRNA plus the shRNA-resistant FGF14b$^{WT}$ (shRNA/FGF14$^{WT}$). The number of neurons tested, $N$, is in parentheses. Inset shows schematic of voltage protocol. (**C**) Exemplar normalized Na$_V$ current traces elicited with a step depolarization to 0 mV from a holding potential of −90 mV and an exemplar single exponential fit (for Scrambled) for the time constant (τ) of inactivation (dotted line). (**D**) τ of inactivation at the indicated test voltages. The number of neurons tested, $N$, is in parentheses. **p < 0.01. (**E**) Exemplar normalized TTX-sensitive late Na$_V$ currents at −20 mV (measured at 150 ms, arrowhead) from a holding potential of −90 mV. (**F**) Amplitude of late Na$_V$ current as a % of peak (transient) Na$_V$ current. The number of neurons tested, $N$, is in parentheses. **p < 0.01. (**G**) Voltage-clamp protocol and exemplar TTX-sensitive resurgent Na$_V$ currents recorded from cerebellar Purkinje neurons transfected with Scrambled control shRNA. The transient current has been clipped. (**H**) Current–voltage relationship of Na$_V$ resurgent currents. The number of neurons tested, $N$, is in parentheses. **p < 0.01. (**I**) Overlay of Na$_V$ resurgent currents for the indicated conditions recorded with

*Figure 2. Continued on next page*

*Figure 2. Continued*

the indicated voltage protocol. The transient current has been clipped. (**J**) Ratio of peak $Na_V$ resurgent current (at +20 mV) to transient $Na_V$ current (at −10 mV). The number of neurons tested, *N*, is in parentheses. **p < 0.01. (**K**) Time to peak of $Na_V$ resurgent current over a broad range of voltages. The number of neurons tested, *N*, is in parentheses. **p < 0.01.

specific effects on $Na_V$ currents (*Lou et al., 2005*; *Rush et al., 2006*), including the regulation of inactivation kinetics (*Laezza et al., 2009*; *Dover et al., 2010*), but the role of the FGF14 N-termini have not been investigated in cerebellar Purkinje neurons. FGF14b is the predominant FGF14 splice variant expressed in brain (*Wang et al., 2000*, *2011*) and the FGF14 splice variant found in the cytoplasm; FGF14a is localized to the nucleus (*Wang et al., 2000*). To test the specific roles of the FGF14b N-terminus, we expressed a FGF14 in which the N-terminus was deleted (FGF14$\Delta^{NT}$). Because the interaction site for the $Na_V$ CTD on FHFs lies within the conserved core domain (*Wang et al., 2012*), deletion of the N-terminus does not affect interaction with the $Na_V$ CTD. Thus, the FGF14$\Delta^{NT}$ exerts a dominant negative effect by competing with endogenous FGF14 for interaction with the $Na_V$ CTD (*Figure 3A*). Current density (*Figure 3B*) and the $V_{1/2}$ of activation (*Figure 3C* and *Table 1*) were unaffected. However, expression of FGF14$\Delta^{NT}$ shifted the $V_{1/2}$ of steady-state inactivation about −10 mV (*Figure 3C*), accelerated kinetics of inactivation (*Figure 3D*), reduced the late $Na_V$ current (*Figure 3E*), and diminished resurgent current (*Figure 3F*) to a degree similar to or even greater than shRNA knockdown. FGF14$\Delta^{NT}$ diminished resurgent $Na_V$ current even more effectively than shRNA knockdown of FGF14. As a control, we overexpressed the FGF14b$^{WT}$ that had rescued the parameters affected by shRNA knockdown (*Figure 1*). Overexpression of FGF14b$^{WT}$ here was without effect (*Figure 3* and *Table 1*). Not only did this serve as a control for FGF14$\Delta^{NT}$, but these data suggest that endogenous FGF14b is saturating. These data also suggest that FGF14b's main regulatory component for Purkinje neuron $Na_V$ currents lies within the FGF14b N-terminus.

The dominant negative effect exerted by FGF14$\Delta^{NT}$ as it replaced the endogenous FGF14 on the $Na_V$ CTD implied that although FGF14 regulates $Na_V$ current through its N-terminus, it may require that FGF14 be in direct interaction with the channel. To test this hypothesis specifically, we designed a FGF14b that was unable to bind to the channel CTD and examined whether expression of this mutant could restore the diminished resurgent current resulting from shRNA knockdown of FGF14. Structural determination of a ternary complex containing FGF13, the $Na_V$1.5 CTD, and calmodulin had identified key amino acids on the conserved FHF interaction surface that participated in the interaction with the conserved $Na_V$ CTD surface (*Wang et al., 2012*). We focused on a highly conserved Arg in the FHF core domain (Arg57 in FGF13U, equivalent to Arg117 in FGF14b), which inserts into a deep pocket on the $Na_V$1.5 CTD binding surface (*Figure 4A*). We therefore generated the homologous R117A mutant (FGF14b$^{R/A}$) in the FGF14b shRNA-resistant cDNA background and tested whether FGF14b$^{R/A}$ could bind to the intact $Na_V$1.6. Although immunoprecipitation of FGF14b$^{WT}$ from lysates of HEK293 cells transfected with FGF14b$^{WT}$ and $Na_V$1.6 showed co-immunoprecipitation of $Na_V$1.6, the mutant FGF14b$^{R/A}$ was unable to co-immunoprecipitate $Na_V$1.6 (*Figure 4B*). Thus, mutation of R117A in FGF14b prevents interaction with $Na_V$1.6. We therefore transfected the shRNA-resistant FGF14b$^{R/A}$ (or the shRNA-resistant FGF14b$^{WT}$) along with the FGF14 shRNA into cultured cerebellar Purkinje neurons to determine the effect of FGF14 interaction with $Na_V$1.6 on $Na_V$ currents. Expression of FGF14b$^{R/A}$ after shRNA knockdown of endogenous FGF14 had no effect on $Na_V$ current density (*Figure 4C*). In contrast to the rescue afforded by FGF14b$^{WT}$, FGF14b$^{R/A}$ did not rescue the key parameters that lead to increased inactivation observed after FGF14 shRNA. $Na_V$ late current and the time constant of inactivation were not different from shRNA (*Figure 4D–F*). Consistent with these findings, we observed the expected reduction in resurgent $Na_V$ current (*Figure 4G–H*). Thus, the FGF14 N-terminus must exert its key effects through FGF14's direct interaction with the $Na_V$ C-terminus.

Because the absence of the FGF14 or its N-terminus accelerated $Na_V$ channel inactivation, diminished late current, and decreased resurgent current, we anticipated that the resultant bias towards $Na_V$ inactivation would adversely affect excitability in cerebellar Purkinje neurons. We tested this hypothesis by determining the effects of the FGF14 and its N-terminus on action potential threshold and on the number of action potentials evoked with depolarizing current injections in current-clamp mode. Under our culture conditions (≤14 days in vitro), we found that 8–10% of cerebellar Purkinje

**Table 1.** $Na_V$ current activation and inactivation parameters in cerebellar Purkinje neurons

| | Activation | | | Inactivation | | |
|---|---|---|---|---|---|---|
| | $V_{1/2}$ (mV) | $K$ | n | $V_{1/2}$ (mV) | $K$ | n |
| Scrambled | −34.4 ± 1.5 | 4.0 ± 0.3 | 21 | −49.7 ± 1.2 | 4.2 ± 0.1 | 19 |
| shRNA | −35.0 ± 1.9 | 4.3 ± 0.5 | 22 | −59.0 ± 1.2** | 4.7 ± 0.2 | 20 |
| shRNA/FGF14b$^{WT}$ | −35.5 ± 1.5 | 3.9 ± 0.3 | 14 | −52.0 ± 0.9 | 5.2 ± 0.3 | 9 |
| shRNA/FGF14$^{RA}$ | −33.2 ± 1.5 | 3.5 ± 0.2 | 12 | −50.9 ± 1.4 | 4.4 ± 0.3 | 11 |
| FGF14b$^{WT}$ | −33.1 ± 1.4 | 4.5 ± 0.5 | 14 | −50.6 ± 1.3 | 4.9 ± 0.3 | 10 |
| FGF14Δ$^{NT}$ | −33.9 ± 2.0 | 3.9 ± 0.5 | 14 | −59.1 ± 0.8** | 4.7 ± 0.3 | 15 |

Mean ± s.e.m. (n), **$p < 0.01$ compared to Scrambled control.

neurons exhibit spontaneous action potentials (firing rate 5–20 Hz) with regular and irregular patterns. When neurons were cultured for extended periods (>20 days in vitro), we observed that ~50% of Purkinje neurons spontaneously fired—more similar to acutely isolated Purkinje neurons (*Raman and Bean, 1999*) and cultured Purkinje neurons (*Gruol and Franklin, 1987*). These data suggest that spontaneous activity depends upon the maturity of the neurons in culture. Because of the difficulty in achieving voltage clamp on Purkinje neurons ≥14 days in vitro, our voltage-clamp experiments were performed on the younger cells (*Figures 2–4*). Thus, we continued with immature neurons for these experiments, and focused on measuring evoked action potentials rather than spontaneous action potentials. Single action potentials were evoked by a 10-ms current injection with 5-pA increments to determine the current threshold to initiate an action potential. *Figure 5A–B* and *Table 2* show that FGF14 knockdown and expression of the dominant negative FGF14bΔ$^{NT}$ markedly increased the current threshold to produce an action potential; other action potential parameters were not affected. Rescue of FGF14 knockdown by co-expression of FGF14b$^{WT}$ restored the current threshold to control levels. Rescue with the non-interacting FGF14b$^{R/A}$ did not. We also analyzed the number of actions evoked during 600 ms current injection over a wide range of current amplitudes. The number of action potentials evoked was significantly larger in control (scrambled shRNA) neurons than in neurons transfected with FGF14 shRNA. Rescue with FGF14b$^{WT}$, but not FGF14b$^{R/A}$, restored the number of evoked action potentials to control levels. Expression of the dominant negative FGF14bΔ$^{NT}$ reduced the number of evoked action potentials to the level observed after FGF14 knockdown (*Figure 5C–D*). Thus, as observed for the individual $Na_V$ channel kinetic parameters, FGF14b, and its N-terminus specifically, is a critical regulator of neuronal firing and intrinsic membrane properties.

## Discussion

Although *FGF14* haploinsufficiency is associated with spinocerebellar ataxia (*Dalski et al., 2005*) and intrinsic excitability of cerebellar Purkinje neurons is reduced in *Fgf14$^{−/−}$* mice (*Shakkottai et al., 2009*), the detailed multifactorial molecular mechanisms by which FGF14 affects neuronal excitability were not previously known. We had found that FGF14 knockdown in cerebellar granule neurons reduced presynaptic voltage-gated $Ca^{2+}$ currents and thereby affected neurotransmission at the cerebellar granule neuron to Purkinje neuron synapse (*Yan et al., 2013*). Here, we found that FGF14b in cerebellar Purkinje neurons affects multiple kinetic parameters of $Na_V$ channels, thereby reducing the resurgent $Na_V$ current that underlies repetitive firing. Together, the accelerated inactivation, reduced channel availability, and decreased late current observed after FGF14 knockdown bias $Na_V$ channels to the inactivated state. Thus, the blocking particle that competes with the inactivation gate for open $Na_V$ channels is disadvantaged. Consequently, the resurgent current, and the resulting ability to support repetitive firing in Purkinje neurons, is reduced as we observed. Thus, our data show that an essential regulatory feature of FGF14 in cerebellar Purkinje neurons is to slow $Na_V$ channel inactivation in order to foster resurgent current.

Resurgent current results from the actions on $Na_V$1.6 of a putative blocking particle, which is thought to be the cytoplasmic C-terminal peptide of the $Na_V$ auxiliary $Na_V$β4 subunit (*Grieco et al., 2005*).

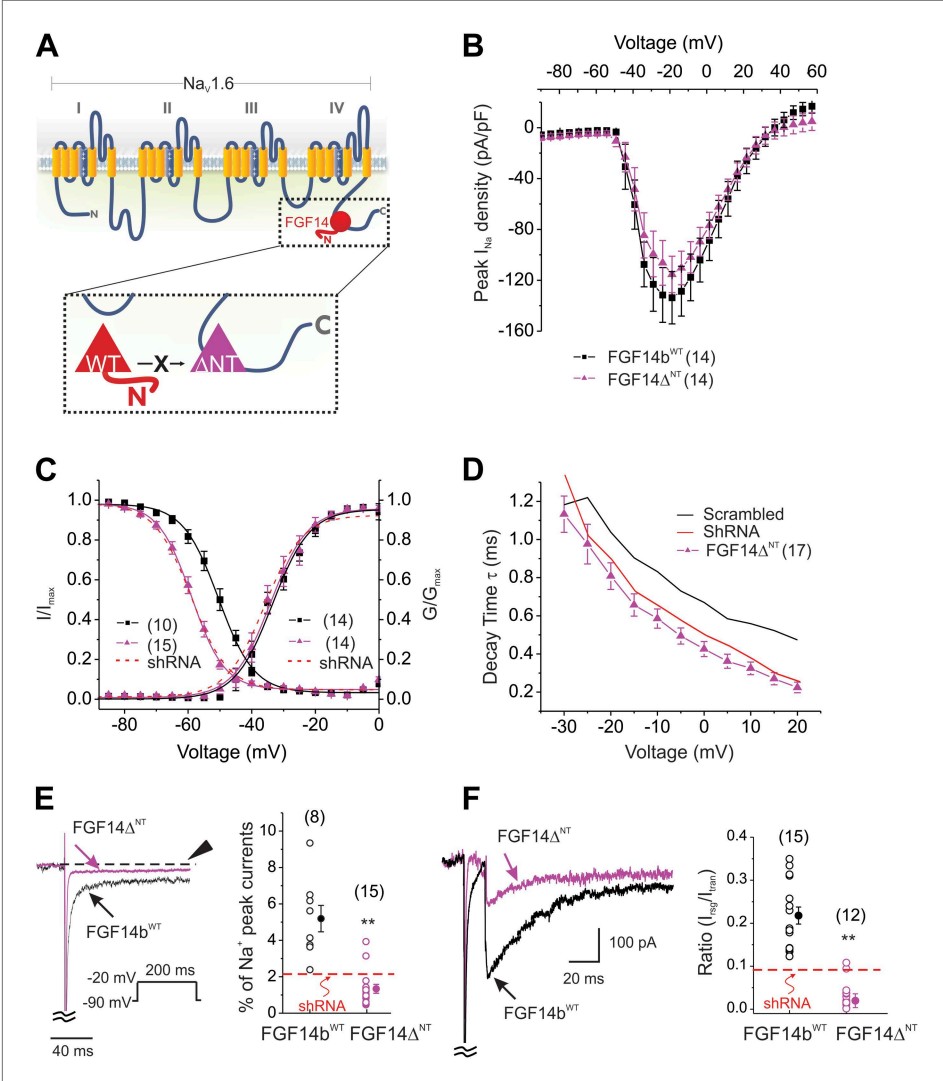

**Figure 3**. Expression of the dominant negative FGF14Δ$^{NT}$ affects multiple Na$_V$ channel biophysical properties indicating essential roles for the FGF14 N-terminus. (**A**) Schematic of endogenous wild type (WT) FGF14 binding to the C-terminus of the Na$_V$1.6 α subunit (top) and magnified schematic of the expressed FGF14Δ$^{NT}$ preventing binding of the WT FGF14 (in box). (**B**) Current–voltage relationships (normalized to cell capacitance) of transient Na$_V$ currents from cerebellar Purkinje neurons transfected with FGF14b$^{WT}$ or FGF14Δ$^{NT}$. The number of neurons tested, *N*, is in parentheses. The current–voltage relationship for Scrambled control shRNA (from **Figure 1**) is shown for comparison. (**C**) Voltage dependence of Na$_V$ channel activation and steady-state inactivation in cerebellar Purkinje neurons transfected with FGF14b$^{WT}$ or FGF14Δ$^{NT}$. The number of neurons tested, *N*, is in parentheses. The curves for Scrambled control shRNA (from **Figure 1**) are shown for comparison. (**D**) τ of inactivation at the indicated test voltages. The number of neurons tested, *N*, is in parentheses. (**E**) Exemplar normalized TTX-sensitive late Na$_V$ currents at −20 mV (measured at 150 ms, arrowhead) from a holding potential of −90 mV. The number of neurons tested, *N*, is in parentheses. **p < 0.01. (**F**) Overlay of Na$_V$ resurgent currents recorded from cerebellar Purkinje neurons transfected with FGF14Δ$^{NT}$ or FGF14b$^{WT}$ and ratio of peak Na$_V$ resurgent current (at +20 mV) to transient Na$_V$ current (at −10 mV). The number of neurons tested, *N*, is in parentheses. The average for FGF14 shRNA knockdown (**Figure 1**) is shown for comparison. **p < 0.01.

Nevertheless, the actions of Na$_V$β4 are not sufficient to generate resurgent Na$_V$ current (**Chen et al., 2008**; **Aman et al., 2009**), suggesting that other components of the Na$_V$ channel complex may be necessary for proper regulation of resurgent Na$_V$ current in cerebellar Purkinje neurons. Here, our knockdown experiments and expression of the dominant negative FGF14bΔ$^{NT}$ show that FGF14b, and

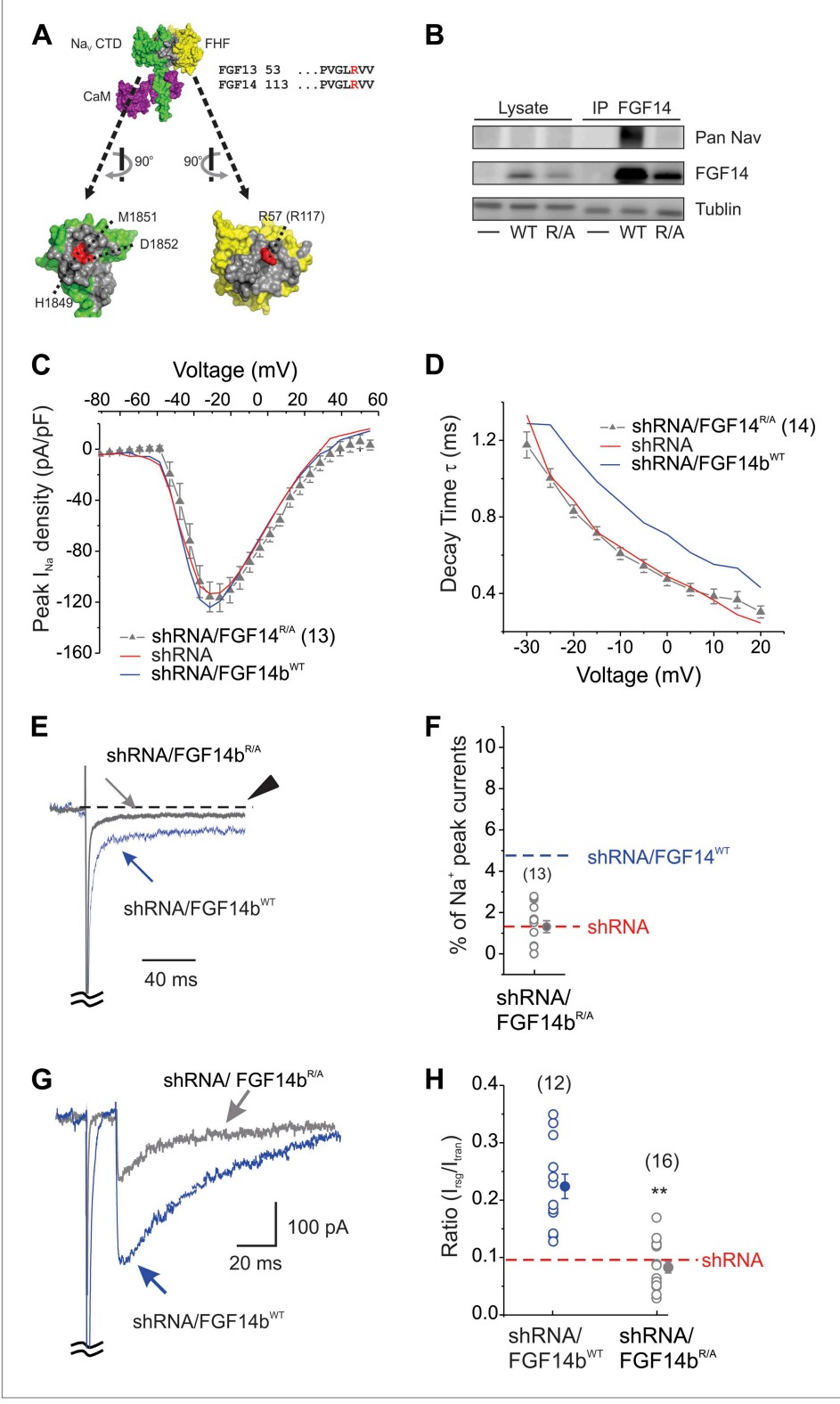

**Figure 4**. The FGF14b[R/A] mutant that prevents interaction with the Na$_V$ C-terminus cannot rescue Na$_V$ kinetic effects of FGF14 knockdown. (**A**) Surface representation of the crystal structure of a ternary complex containing the Na$_V$1.5 C-terminus domain (CTD, green), FGF13 (yellow), and calmodulin (purple); the interaction surfaces are colored gray (PDB ID: 4DCK). The critical R57 in FGF13 (equivalent to R117 in FGF14) is indicated in red, as is its binding pocket

*Figure 4. Continued on next page*

*Figure 4. Continued*

on the $Na_V1.5$ CTD. (**B**) Co-immunoprecipitation (IP) of $Na_V1.6$ and $FGF14b^{WT}$ or $FGF14b^{R/A}$ expressed in HEK293 cells, showing that the $FGF14b^{R/A}$ is unable to interact with the intact $Na_V1.6$. Immunoblots (IB) were performed with the indicated antibodies. (**C**) Current–voltage relationship (normalized to cell capacitance) of transient $Na_V$ currents from cerebellar Purkinje neurons transfected with $FGF14b^{R/A}$. The number of neurons tested, $N$, is in parentheses. The current–voltage relationship for FGF14 knockdown (shRNA) and knockdown rescued with shRNA-insensitive $FGF14b^{WT}$ (shRNA/$FGF14b^{WT}$) from *Figure 1* are shown for comparison. (**D**) $\tau$ of inactivation at the indicated test voltages. The number of neurons tested, $N$, is in parentheses. (**E**) Exemplar normalized TTX-sensitive late $Na_V$ currents at −20 mV (measured at 150 ms, arrowhead) from a holding potential of −90 mV and (**F**) amplitude of late $Na_V$ current as a % of peak (transient) $Na_V$ current. The number of neurons tested, $N$, is in parentheses. Averages for shRNA and for shRNA/$FGF14b^{WT}$ (see *Figure 1*) are shown for comparison. (**G**) Overlay of $Na_V$ resurgent currents recorded from cerebellar Purkinje neurons transfected with $FGF14b^{R/A}$ or $FGF14b^{WT}$ and (**H**) ratio of peak $Na_V$ resurgent current (at +20 mV) to transient $Na_V$ current (at −10 mV). The number of neurons tested, $N$, is in parentheses. The average for FGF14 shRNA knockdown (*Figure 1*) is shown for comparison. \*\*p < 0.01.

specifically its N-terminus, strongly influence resurgent $Na_V$ current. FGF14 knockdown caused a ~60% reduction. Expression of FGF14$\Delta^{NT}$ was even more efficient, resulting in a ~85% reduction, perhaps because overexpression of FGF14$\Delta^{NT}$ was more efficient in replacing endogenous FGF14b than shRNA knockdown was in eliminating endogenous FGF14b. Thus, FGF14 may cooperate with $Na_V\beta4$ to generate resurgent current, perhaps by biasing $Na_V1.6$ channels towards inactivation and thereby disadvantaging the blocking particle. Our results examining effects on total $Na_V$ current must be assessed in the context of FGF14 effects not only on $Na_V1.6$ channels but also on $Na_V1.1$ channel, the other major source of $Na_V$ currents in cerebellar Purkinje neurons (*Kalume et al., 2007*). Those $Na_V$ channels are also influence by FGF14 (*Lou et al., 2005*). Thus, the effects on resurgent current that we attribute to FGF14 may result from FGF14's overall influence of $Na_V$ channel kinetics within Purkinje neurons.

Our results may also provide some insight into the more variable phenotypes and decreased penetrance in spinocerebellar ataxia patients with *FGF14* haploinsufficiency (*Dalski et al., 2005*; *Coebergh et al., 2013*) vs patients with a *FGF14* mutation ($FGF14b^{F150S}$) found in a large Dutch spinocerebellar ataxia kindred (*van Swieten et al., 2003*). Studies in hippocampal neurons suggest that the $FGF14b^{F150S}$ mutant exerts a dominant negative mechanism in which the axon initial segment is depleted of $Na_V$ channels and $Na_V$ current density is reduced (*Laezza et al., 2007*). Here, we found no effect on $Na_V$ current density in cerebellar Purkinje neurons after FGF14 knockdown. A reduction in $Na_V$ current density (with the $FGF14b^{F150S}$ mutation) compared to an effect solely on $Na_V$ channel kinetics (with FGF14 haploinsufficiency) may explain the higher penetrance and decreased phenotypic variability in patients with the $FGF14b^{F150S}$ mutation.

An important aspect of this study examining effects on $Na_V$ channel kinetics was that our analyses were performed in Purkinje neurons. Studying FHFs in their native cellular context appears to be crucial for defining their specific roles. Previous investigations of FGF14-dependent regulation of neuronal $Na_V$ channels in heterologous expression systems led to different conclusions from experiments in neurons. For example, expression of $Na_V1.6$ and $FGF14b^{WT}$ in ND7/23 cells almost eliminated $Na_V$ current density (*Laezza et al., 2009*) while overexpression of $FGF14b^{WT}$ in hippocampal neurons increased $Na_V$ current density (*Laezza et al., 2007*). Similarly, expression of FGF14$\Delta^{NT}$ in ND7/23 cells increased current density, while here we observed that FGF14$\Delta^{NT}$ expression in cerebellar Purkinje neurons exerted dominant-negative effects upon $Na_V$ channel kinetics, but no consequences for $Na_V$ current density. While the etiology of the different results in neurons compared to heterologous expression systems is not known, we speculate that additional regulatory factors present in neurons are necessary for physiologic regulation of $Na_V$ channels by FHFs. Indeed, we have observed similarly disparate results for the related FGF13 and the cardiac $Na_V1.5$ channel. In HEK293 cells, FGF13 co-expression reduces $Na_V1.5$ current density (not shown), but we showed that the role FGF13 in cardiomyocytes is to increase $Na_V$ current density (*Wang et al., 2011*; *Hennessey et al., 2013*).

Finally, our data also provide information about possible consequences of *FGF14* overexpression, hypothesized as pathologic in rare cases. Some patients with spinocerebellar ataxia, but not controls, harbor synonymous variants in *FGF14*. These variants encode a more frequently used codon than the

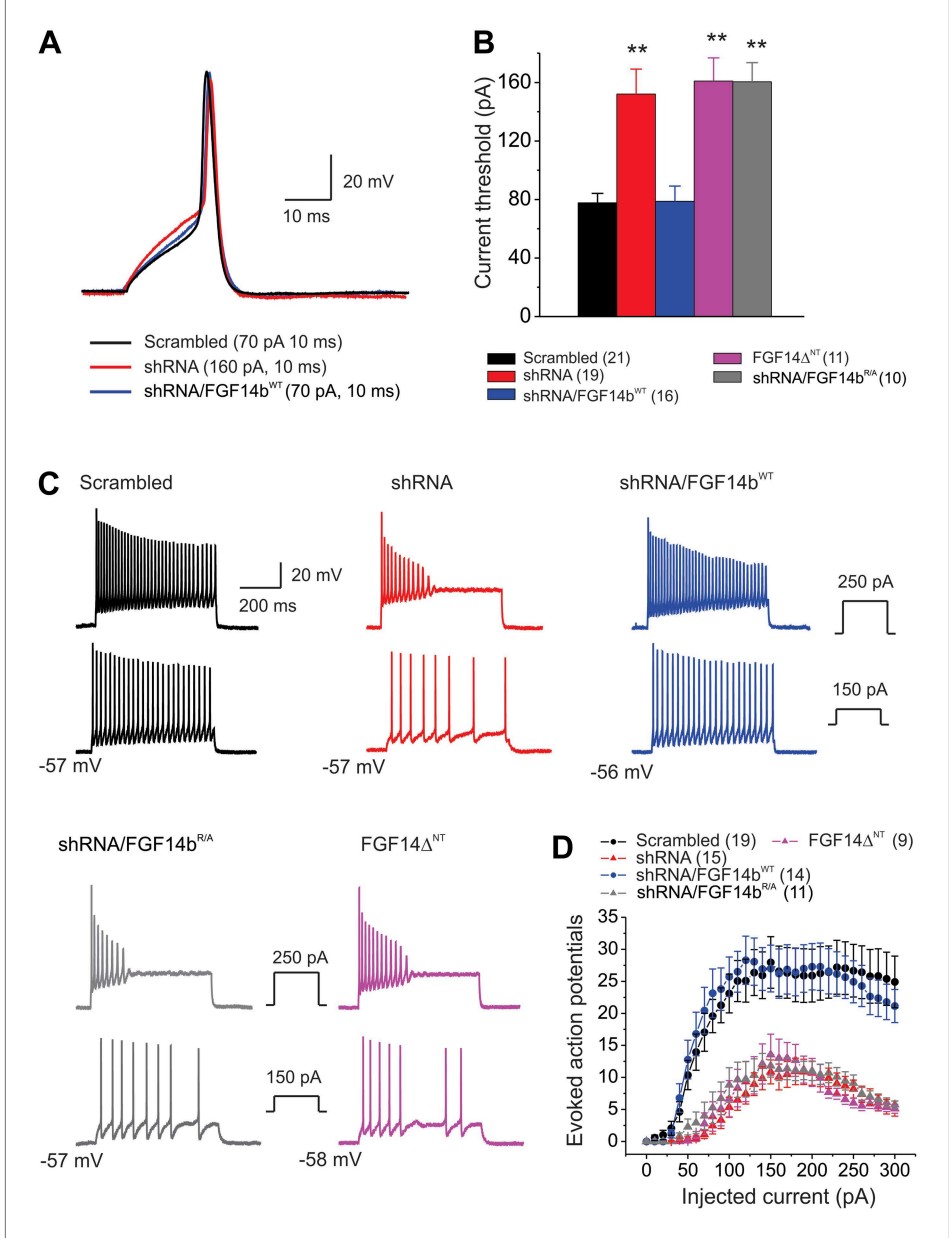

**Figure 5**. Action potential dynamics and repetitive spiking are dependent upon FGF14 and its N-terminus. (**A**) Overlay of single action potentials evoked with a 10 ms current injection (current amplitude shown in parentheses). (**B**) Current threshold to induce action potentials for the indicated conditions. **p < 0.01. The number of neurons tested, *N*, is in parentheses. Additional summary data are presented in *Table 2*. (**C**) Example evoked action potentials for the indicated treatments at two separate current injection amplitudes (shown in insets). The resting membrane potential is indicated. (**D**) The number of evoked action potentials for the indicated amplitude of current injection. The number of neurons tested, *N*, is in parentheses.

wild type sequence, leading to the hypothesis that patients with these variants would overexpress *FGF14* as a cause of disease (*Dalski et al., 2005*). We found, however, that overexpression of FGF14b$^{WT}$ did not affect current density or any of the measured Na$_V$ kinetic properties (*Figure 3* and *Table 1*). Thus, if these variants identified in spinocerebellar ataxia patients are truly associated with disease, our data suggest mechanisms other than FGF14 overexpression.

In summary, these data add to a growing appreciation of physiologic and pathophysiologic roles for FHFs in neurons. In the case of spinocerebellar ataxia, our data suggest that FGF14 is a critical

**Table 2.** Intrinsic membrane properties and single action potential (AP) characteristics measured in cerebellar Purkinje neurons

| | Scrambled | shRNA | shRNA/FGF14b^WT | shRNA/FGF14b^RA | FGF14Δ^NT |
|---|---|---|---|---|---|
| Input resistance (MΩ) | 221.8 ± 15.8 (7) | 238.0 ± 49.5 (11) | 266.6 ± 24.5 (11) | 225.9 ± 10.8 (6) | 269.5 ± 30.2 (6) |
| Resting membrane potential (mV) | −54.5 ± 1.3 (21) | −55.8 ± 1.2 (19) | −54.4 ± 1.0 (16) | −55.4 ± 0.7 (10) | −54.9 ± 1.4 (11) |
| Current threshold (pA) | 77.8 ± 6.4 (21) | 142.1 ± 18.5 (19)** | 78.8 ± 10.4 (16) | 170.5 ± 16.7 (10)** | 191.0 ± 22.1 (11)** |
| AP threshold (mV) | −35.6 ± 1.0 (21) | −32.8 ± 1.2 (19) | −34.5 ± 0.8 (16) | −32.8 ± 1.4 (10) | −32.7 ± 0.8 (11) |
| AP amplitude (mV) | 78.8 ± 3.2 (21) | 77.5 ± 2.8 (19) | 78.9 ± 4.5 (16) | 79.1 ± 4.3 (10) | 73.1 ± 4.7 (11) |
| AP duration (ms) | 2.3 ± 0.2 (21) | 2.0 ± 0.1 (19) | 2.2 ± 0.1 (16) | 1.8 ± 0.1 (10) | 2.1 ± 0.3 (11) |

Mean ± s.e.m. (n), **$p < 0.01$ compared to Scrambled control.

regulator of the ability of cerebellar Purkinje neuron $Na_v$ channels to support resurgent $Na_v$ current and thereby allow Purkinje neurons to fire repetitively. Thus, we provide a molecular understanding for the observation that FGF14 regulates Purkinje neuron intrinsic excitability (*Shakkottai et al., 2009*). In combination with our data showing that FGF14 also regulates presynaptic voltage-gated $Ca^{2+}$ channels at the cerebellar granule cell to Purkinje cell synapse, these data pinpoint several specific mechanisms by which mutations in *FGF14* underlie spinocerebellar ataxia.

# Materials and methods

## Primary cerebellar neuron culture and transfection

This study was performed in strict accordance with the recommendations in the Guide for the Care and Use of Laboratory Animals of the National Institutes of Health. All of the animals were handled according to approved Institutional Animal Care and Use Committee (IACUC) of Duke University (protocol #A292-13-11). Primary dissociated cerebellar neurons were cultured as previously described (*Yan et al., 2013*) with some modifications. Briefly, primary cultures were prepared from P6-P8 C57BL6 mice. Cerebella were excised, dissected on ice, digested with 0.25% trypsin for 10 min at 37°C with Dulbecco's Modified Eagle's Medium (DMEM, Sigma, St. Louis, MO), and dissociated into single cells by gentle trituration. The cells were seeded onto coverslips coated with 50 µg/ml poly-D-lysine (Sigma) and 25 µg/ml laminin (Sigma) at a density of $2.5–3.0 × 10^5$ cells/coverslip in DMEM supplemented with 10% heat-inactivated fetal bovine serum (FBS). The neurons were maintained in a humidified incubator in 5% $CO_2$ at 37°C. After 15–16 hr, the medium was replaced with Basal Medium Eagle (BME, Sigma) supplemented with 2% B27 (Invitrogen), 5% FBS, 25 µM uridine, 70 µM 5-fluorodeoxyuridine, and 20 mM KCl. Neurons were transiently transfected at 4 days in vitro with 1 µg plasmid DNA and/or shRNA plasmids per coverslip with calcium phosphate. Experiments were performed 8–10 days after transfection (12–14 days in vitro). To achieve good voltage control, all recordings were performed in neurons cultured no longer than 14 days in vitro.

## shRNA and cDNA construction

shRNAs targeted to FGF14 were designed with Invitrogen's RNAi Designer as previously described (*Yan et al., 2013*) and the sequences were cloned into pLVTHM (Addgene). After evaluating several candidates, we found that the most effective shRNA sequence was 5′ CGCGTGGA GGCAAACCAGTCAACAAGTGCATTCAAGAGATGCACTTGTTGACTGGTTTGC CTCCTTTTTTAT 3′, which was used for the experiments described here. The scrambled shRNA which bears no homology to genes in the rodent genome (*Wang et al., 2011*) has the sequence: 5′-CGCGTGACCC TTAGTTTATACCTATTCAAGAGATAGGTATAAACTAAGGGTCTTTTTTAT-3′. FGF14 rescue and overexpression experiments were performed with FGF14 constructs (either full-length or with the N-terminus truncated) cloned into pcDNA3.1 also containing tdTomato. Mutagenesis to obtain FGF14b^{R/A} was performed using QuikChange (Agilent). All plasmids were verified by sequencing.

## Electrophysiological recordings

Purkinje neurons were identified by their characteristic teardrop morphology and large size. Whole-cell $Na_v$ currents and membrane voltage were recorded at 23–25°C using an EPC 10 USB patch

amplifier (HEKA Elektronik). The signal was filtered at 2.9 Hz and digitized at 20 Hz. $Na_V$ currents were recorded with an extracellular solution containing (in mM) 124 NaCl, 20 TEA-Cl (tetraethyl-ammonium Chloride), 5 HEPES (4-(2-hydroxyethyl)-1-piperazineethanesulfonic acid), 10 glucose, 0.3 $CdCl_2$, 2 $BaCl_2$, and 4-AP (4-aminopyridine). NaOH was added to achieve pH 7.3 (300–310 mOsm). Borosilicate glass patch pipettes (resistances within 3–4 MΩ) were filled with the following internal solution (in mM): 125 CsF, 10 NaCl, 10 HEPES, 15 TEA-Cl, 1.1 ethylene glycol tetraacetic acid (EGTA), and 0.5 Na-guanosine-5′-triphosphate (Na-GTP), pH adjusted to 7.3 with CsOH (290–300 mOsm). Tetrodotoxin (TTX)-sensitive $Na_V$ currents were isolated by subtraction of recordings performed in 1 µM TTX from the control recordings. Series resistance was compensated >70%. Current-clamp recordings were performed after obtaining seal resistances >1.2 GΩ. The internal solution was (in mM) 130 K-gluconate, 5 NaCl, 2 $MgCl_2$, 10 HEPES, 4 Mg-ATP, 0.5 Na-GTP, 0.2 EGTA, and 10 phosphocreatine. KOH was used to obtain pH 7.3 (290–300 mOsm). The external solution was (in mM) 140 NaCl, 5 KCl, 2 $CaCl_2$, 1 $MgCl_2$, 5 HEPES, and 10 glucose, pH adjusted to 7.3 with NaOH. Resting membrane potential was directly measured in current-clamp mode after membrane rupture and only cells with a resting membrane potential more negative than −50 mV were studied. The liquid junction potential was not corrected. Input resistance was determined from membrane voltage deflection, evoked by 600-ms hyperpolarizing current injections (0 to −300 pA in steps of 50 pA) and calculated from the measured slope. Single action potentials were elicited by 10 ms depolarizing current injections with 5-pA increments. All drugs were from Sigma Aldrich, except for TTX (Abcam Biochemicals).

## Protocols and data analysis

Data analysis was performed using FitMaster (HEKA), Excel (Microsoft), and Origin software. All averaged data are presented as mean ± SEM. Statistical significance was determined using Student's *t* or one-way ANOVA tests. Calculated p values of ≤0.05 were accepted as evidence of statistically significant differences. For current amplitude, neurons were held at −90 mV and transient $Na_V$ current was elicited by depolarizing pulses of 20 ms from −90 mV to +55 mV in 5-mV increments. Current density was obtained by normalizing peak $Na_V$ current to membrane capacitance. Resurgent $Na_V$ current was evoked with repolarizing steps from +20 mV to a range of voltages between −80 mV and +5 mV in 5-mV increments for a 200-ms test pulse after 15 ms conditioning steps. $Na_V$ activation curves were obtained by transforming current data to conductance (G), with the equation $G = I_{Na}/(E_m − E_{rev})$, where: $I_{Na}$ is the peak current; $E_m$ is the membrane potential; and $E_{rev}$ is the reversal potential of $I_{Na}$; and fitted with a Boltzmann equation of the form: $G = G_{max}/[1 + exp(V_{1/2} − V)/k]$, where $G_{max}$ is the extrapolated maximum $Na^+$ conductance, V is the test voltage, $V_{1/2}$ is the half-activation voltage, and *k* is the slope factor. For $Na_V$ steady-state inactivation, a voltage step to −20 mV for 20 ms was applied from a holding potential of −90 mV to preferentially activate $I_{Na}$ after pre-pulse conditioning voltage steps of 300 ms in duration (ranging from −90 to +20 mV) in 5-mV increments. Steady-state inactivation curves were constructed by plotting the normalized peak current amplitude elicited during the test pulse as a function of the conditioning pre-pulse. A Boltzmann relationship, $I/I_{max} = (1 + exp((V − V_{1/2})/k))^{−1}$ was used to fit the data where $I_{max}$ is current elicited by the test pulse after a −90-mV prepulse; $V_{1/2}$ is half-inactivation voltage; *k* is the slope. Late $Na_V$ current was measured at 150 ms during a 200 ms depolarizing pulse to −20 mV. The rate of decay of the transient $Na_V$ current (τ) was obtained from a single exponential function, $I_{(t)} = I_{Na} exp^{(−t/τ)} + I_{SS}$, where $I_{(t)}$ is the amplitude of the current at time *t*; $I_{SS}$ is the steady-state current during a single voltage step.

In current clamp, action potential amplitude was measured as the peak voltage with respect to the baseline 10 ms before the peak of the action potential. The action potential duration was evaluated at half-amplitude. Evoked action potentials were elicited by injecting a depolarizing current from 0 pA to 300 pA for 600 ms duration with 10-pA increments.

## Immunoprecipitation and Western blotting

Cerebella were isolated from two 5-week-old C57BL/6J mice and homogenized with a mortar and pestle in 8 ml of lysis buffer composed of 150 mM NaCl, 50 mM Tris-Cl, 1% Triton, and 1% sodium deoxycholate. Lysate was incubated on ice for 2 hr, passed through a 20g needle 20 times, and rocked end over end for 30 min at 4°C. The lysate was cleared of insoluble material by centrifugation at 4000 rcf at 4°C for 30 min followed by a second centrifugation at 17,100 rcf at 4°C

for 10 min. The lysate was preincubated with 30 µl of Protein A/G (Santa Cruz Biotechnology) beads for 2 hr at 4°C. The beads were gently spun down at 0.4 rcf for 1 min to separate them from the lysate, which was then incubated with 20 µg of either anti-FGF14 (NeuroMab) or non-immune mouse IgG (Santa Cruz Biotechnology) overnight at 4°C. In order to immunoprecipitate protein complexes, 30 µl of Protein A/G beads were added and incubated with the lysate for 2 hr. Beads were then washed three times with lysis buffer followed by elution in 2× LDS Sample Buffer (Novex NuPage) plus 10 mM dithiothreiotol. Protein samples were run on 8–16% Tris-glycine SDS page gels and transferred to PVDF membranes. Primary antibodies used were rabbit polyclonal anti $Na_v$1.6 from Millipore (1:200), anti-β-tubulin from Sigma (1:1000) as a loading control, and rabbit polyclonal anti-FGF14 (1:200), which was a kind gift from the Ornitz lab (Washington University, St. Louis). HEK293T cells on 100-mm plates were transfected with Lipofectamine 2000 (Invitrogen) when cells were at ~60% confluency with 9 µg of $Na_v$1.6 and 3 µg of FGF14b$^{WT}$ or FGF14b$^{R/A}$. Cells were washed with ice-cold PBS 60 hr after transfection, and cell lysates were prepared with the addition of lysis buffer containing 150 mm NaCl, 50 mm Tris–HCl, pH 7.5, 1% Triton with protease inhibitor (Roche). The pelleted cells were pipetted up and down 20 times with lysis buffer, incubated at 4°C for 1 hr and then centrifuged at 16,000×$g$ for 10 min at 4°C. Immunoprecipitation was performed on 2 mg of lysate with 2 µg of anti-FGF14 antibody (Neuromab). The samples were rocked gently at 4°C for 1 hr followed by addition of 30 µl of protein A/G-agarose slurry. The samples were rotated overnight at 4°C and microcentrifuged at 7000 rpm for 2 min. After washing with lysis buffer three times, 40 µl of loading buffer was added to the pellet, and protein was eluted from the beads by heating at 70°C for 20 min. The samples were subjected to NuPAGE 8–16% Bis-Tris gels (Invitrogen). The proteins were transferred to nitrocellulose membranes and subsequently immunoblotted with the anti-FGF14, anti-pan $Na_v$ antibody, and anti-tubulin antibody.

## Acknowledgements

We thank Dr Amanda H Lewis for insightful discussions. This work was supported by NIH R01 HL071165 and R01 HL112918 (GSP).

## Additional information

### Funding

| Funder | Grant reference number | Author |
|---|---|---|
| National Heart, Lung, and Blood Institute | R01 HL071165 | Geoffrey S Pitt |
| National Heart, Lung, and Blood Institute | R01 HL112918 | Geoffrey S Pitt |

The funder had no role in study design, data collection and interpretation, or the decision to submit the work for publication.

### Author contributions

HY, Conception and design, Acquisition of data, Analysis and interpretation of data, Drafting or revising the article; JLP, CW, Acquisition of data, Analysis and interpretation of data, Drafting or revising the article; GSP, Conception and design, Analysis and interpretation of data, Drafting or revising the article

### Author ORCIDs

Geoffrey S Pitt, http://orcid.org/0000-0003-2246-0289

### Ethics

Animal experimentation: This study was performed in strict accordance with the recommendations in the Guide for the Care and Use of Laboratory Animals of the National Institutes of Health. All of the animals were handled according to approved institutional animal care and use committee (IACUC) of Duke University for the protocol #A292-13-11.

# Additional files

## Major dataset

The following previously published dataset was used:

| Author(s) | Year | Dataset title | Dataset ID and/or URL | Database, license, and accessibility information |
|---|---|---|---|---|
| Wang C, Chung BC, Yan H, Lee SY, Pitt GS | 2012 | Crystal structure of the C-terminus of voltage-gated sodium channel in complex with FGF13 and CaM | http://www.pdb.org/pdb/explore/explore.do?structureId=4DCK | Publicly available at RCSB Protein Data Bank. |

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
