## [Decision Letter]

Thank you for sending your work entitled “FGF14 Modulates Resurgent Sodium Current in Mouse Cerebellar Purkinje Neurons” for consideration at *eLife*. Your article has been favorably evaluated by a Senior editor and 3 reviewers as well as Gary Westbrook, a member of our Board of Reviewing Editors.

The Reviewing editor and the other reviewers discussed their comments before we reached this decision, and following comments were assembled to help you prepare a revised submission. As you will see the reviewers thought the data was convincing in showing that the intracellular protein FGF14, which has previously been shown to interact with sodium channels, regulates the ability of the channels to generate “resurgent” sodium current, an unusual kinetic behavior of sodium channels linked to rapid firing, whereby sodium current flows during recovery from inactivation. The experiments are based on combined shRNA and dominant negative protein experiments using a preparation of cultured Purkinje neurons. This is a powerful combination of molecular tools and physiological function. The implication of FGF14 in resurgent current is of considerable significance, especially because mutations affecting FGF14 are implicated in some human spinocerebellar ataxias.

The strength of this manuscript is the clear presentation of the functional effects of FGF-14 on sodium currents in Purkinje neurons in Figures 1, 2, 3, 4 and 5. These studies are not a dramatic conceptual advance, but the analysis presented here is more comprehensive and carefully documented than previous work and significantly increases our understanding of FGF-14 effects. The weakness of this manuscript is the interpretation of the regulatory effects observed in molecular terms. As the authors acknowledge, the molecular components required for resurgent sodium current remain unknown, and the molecular mechanism of generation of resurgent current is not substantially advanced by the work presented here. Specifically, no evidence was presented to support the authors' proposal that FGF14 serves as a blocking particle and thus the discussion of the molecular mechanism throughout the manuscript (Abstract, Introduction, Figure 6 and Discussion) should be removed as commented below.

Major points:

1) These experiments were done in cultured Purkinje neurons but no information is given about (23) the extent to which their activity resembles that of Purkinje neurons in other preparations (acutely isolated, slice, in vivo) (21) the age of the cultures (DIV) or (8) what efforts were made to achieve good voltage control, which is expected to be difficult in a culture preparation with big currents and axons. The culture preparation is necessary to do these experiments, but addressing these points is important because the data suggest that the cultured neurons do not fire spontaneously (Methods, “only cells with a resting membrane potential more negative than -50 mV were studied”; no indication of holding current to keep cells silent; no spikes evident before steps in current clamp records. Note the extracellular solution for recording spikes is also not reported), and if the authors wish to make a case that FGF14 affects Purkinje cell firing the link to the more physiological condition should be made.

2) The resurgent current seems to rise very rapidly (and looks generally different from that in acutely isolated cells) although that might just be the time base. What was the time to peak of the resurgent current? It would be helpful to illustrate the kinetics of resurgent currents a wider range of potentials to use as comparison to acutely isolated cells on which many earlier experiments are done. Even if there are differences between the acute and cultured preparations, it would be worth cataloguing systematically.

3) Abstract:

“These data suggest that the FGF14b N-terminus may serve as a blocking particle and/or cooperate with NaVbeta 4...” and Discussion as noted. There are no data on open channel block in this manuscript and this conclusion is unsupported. The data show nicely that reduction of functional FGF14 favors inactivation. Because any modulation that favors inactivation will reduce resurgent current, the observed reduction in peak resurgent current is predicted, though still worth documenting. The Results are presented quite clearly, but in the second paragraph of the Discussion the authors surprisingly and unnecessarily extrapolate to the suggestion that FGF14 is itself a blocker or forms part of a blocking complex. The existing data have no direct relationship to this conclusion, and actually seem to argue against the need to invoke this idea for the following reasons:

a) Losing FGF14 function speeds current decay at depolarized potentials, rather than slowing current decay at depolarized potentials, which is what would be expected if open channel block is directly disrupted (Khaliq et al. 2003, J Neurosci note also that these were the effects of trypsin and chymotrypsin in Grieco et al. 2005, Neuron, when the blocker was removed by proteolysis, although the authors argue that FGF14 might be a blocker because of trypsin sites).

b) Loss of FGF14 reduces resurgent current but doesn't seem to change its kinetics, which might be expected if the interaction of the blocking particle with the channels is itself altered.

If the authors want to investigate whether FGF14 can block Na channels, they should simply test whether the N-terminus can make a resurgent current in channels that don't have a native blocker. In the absence of some kind of direct evidence of block, the arguments based on whether the protein has lysines or trypsin sites, etc. are unconvincing and irrelevant. Most importantly, the discussion of FGF14 as a blocker is incidental to the line of the paper and could be cut with no loss to the manuscript.

4) Abstract:

a) “Cerebellar Purkinje neurons fire repetitive action potentials because of a voltage-gated Na+ (NaV) ”resurgent“ current” That is maybe overstating the role of resurgent sodium current. There are neurons withour resurgent current that can still fire repetitively. Something like “Rapid firing of cerebellar Purkinje neurons is facilitated by voltage-gated Na+ (NaV) ”resurgent“ current...” would be more accurate.

b) “Resurgent current results from unbinding of a blocking particle that previously terminated Na+ influx through the channel and simultaneously prevented channel inactivation.” This might be clearer as “Resurgent current results from unbinding of a blocking particle that competes with normal channel inactivation.”

5) Introduction:

a) “Cerebellar Purkinje neurons, which provide the sole cerebellar output signal...”. Not true. A correct statement would be “Cerebellar Purkinje neurons, which provide the sole output from the cerebellar cortex....” (not the cerebellum as a whole).

b) “This is an unusual behavior because, after initially opening, NaV channels in most neurons rapidly enter an inactivated state. This diminishes their ability to reopen quickly that defines the repetitive firing seen in Purkinje neurons.” This was a little clumsy. Something like “Fast repetitive firing in Purkinje neurons is promoted by unusual characteristics associated with sodium channel inactivation, whereby channels recover unusually rapidly from inactivation and in doing so pass a 'resurgent' sodium current, which helps drive the cell to fire a subsequent action potential.”

c)“First, more than half of the NaV current in Purkinje neurons is carried by NaV1.6, an isoform that is particularly reluctant to enter the inactivated state (23).” This is a little misleading (because inactivation is still 95-99% complete) . Something like “..an isoform for which inactivation is somewhat less complete compared to others” would be better.

6) “Although this result appears to contrast with a report that resurgent NaV current was unaffected by the absence of FGF14 in cerebellar granule neurons (5), the protocol employed in that study was markedly different. Instead of measuring resurgent current, it evaluated channel recovery from inactivation.”

This is incorrect. Table 1 from Neuron 55(3):449-463 shows statistics for voltage clamp measurements of resurgent sodium current. However, as the measurements were in cerebellar granule cells, there is no real conflict. Also, the measurements reported were for resurgent current at -85 mV, which in the present results Figure 2 showed much less change than at -40 mV where resurgent current is maximal.

7) “FGF14 is expressed in the hippocampus (1), but resurgent NaV current is not observed in hippocampal neurons unless an exogenous blocking particle is added through the patch pipette (8).”

This is true for CA3 pyramidal neurons but not CA1 pyramidal neurons, which do express resurgent sodium current (Royeck M, Horstmann MT, Remy S, Reitze M, Yaari Y, Beck H. Role of axonal NaV1.6 sodium channels in action potential initiation of CA1 pyramidal neurons. J Neurophysiol. 2008 100:2361-80.)

8) Table 1 includes several non-standard or ambiguous measurement parameters that are not defined in Methods, namely “AP rise time”, “AP decay time”, and “AP integral”. Because these do not change and are not referred to, they should be deleted or Methods should define how they were measured. (“AP rise time” is especially ambiguous and hard to interpret, because it is longer than AP duration.)

9) The authors write as though Purkinje neurons express only Nav1.6 channels. In fact, previous work suggests the presence of other sodium channels as well. For example, specific immunostaining and gene knockout indicate that Nav1.1 channels are present and constitute a significant fraction of sodium current in the cell bodies of dissociated Purkinje neurons (Kalume et al., J Neurosci, 2007). The presence of multiple sodium channel types in Purkinje neurons complicates all of the quantitative interpretations of the results presented here because it is unknown whether FHF-14 affects all of the sodium channels types that may be present in the same way.

10) Figure 4. Why are interactions of Nav1.5 and FHF-13 reported here? These results should be replaced with experiments on Nav1.6 and FHF-14, which form the primary focus of this manuscript.

11) Figure 6. The molecular model developed in Figure 6 and the Discussion is highly speculative. There is no experimental basis for the interaction between the Navbeta4 subunit and FGF14 depicted in Figure 6, and indeed the role of Navbeta4 as a blocking protein has yet to be demonstrated for Purkinje cells. Similarly, the authors indicate that FHF-14 also is unable to generate resurgent sodium current when co-expressed with sodium channels, so there is no experimental basis for its proposed role as a blocking particle that generates resurgent sodium current either. If the authors wish to propose this molecular mechanism, they need to show that co-expression of Navbeta4, FHF-14, or both can generate resurgent sodium current in a cell type in which none is observed before co-expression.

12) The authors base much of the impact of their manuscript on elucidating the molecular mechanism of resurgent sodium current. However, as developed in the points above, their work does not directly address this problem. The authors need to re-orient the Abstract and Discussion and provide a more persuasive presentation of the importance of their findings for the physiology of Purkinje neurons and the pathophysiological effects of spinocerebellar ataxia mutations in FHF-14. The discussion of the molecular role of FHF-14 and Navbeta4 should be greatly reduced in favor of consideration of the potential impact of regulation by FHF-14 in the physiological function and pathophysiology of Purkinje neurons.

---

## [Author Response]

*1) These experiments were done in cultured Purkinje neurons but no information is given about (*[23]*) the extent to which their activity resembles that of Purkinje neurons in other preparations (acutely isolated, slice, in vivo) (*[21]*) the age of the cultures (DIV) or (*[8]*) what efforts were made to achieve good voltage control, which is expected to be difficult in a culture preparation with big currents and axons. The culture preparation is necessary to do these experiments, but addressing these points is important because the data suggest that the cultured neurons do not fire spontaneously (Methods, “only cells with a resting membrane potential more negative than -50 mV were studied”; no indication of holding current to keep cells silent; no spikes evident before steps in current clamp records. Note the extracellular solution for recording spikes is also not reported), and if the authors wish to make a case that FGF14 affects Purkinje cell firing the link to the more physiological condition should be made*.

Similar to the activity of Purkinje neurons in acutely isolated slice or *in vivo*, we observed spontaneous firing in our cultured Purkinje neurons when they remained in culture for ≥ 20 days *in vitro*. However, our experiments were performed with 12-14 days *in vitro* Purkinje neurons, allowing us to achieve reasonable voltage control that was less attainable in older cultures. In this context, we chose to study evoked rather than spontaneous action potentials. These culture details have been added to the Methods and this specific limitation is now addressed in the text.

The extracellular solution is now reported (Methods Section).

*2) The resurgent current seems to rise very rapidly (and looks generally different from that in acutely isolated cells) although that might just be the time base. What was the time to peak of the resurgent current? It would be helpful to illustrate the kinetics of resurgent currents a wider range of potentials to use as comparison to acutely isolated cells on which many earlier experiments are done. Even if there are differences between the acute and cultured preparations, it would be worth cataloguing systematically*.

We have calculated the time to peak for resurgent current over a wide range of potentials and have added these data to Figure 2.The kinetics in cultured Purkinje neurons are only slightly faster than those reported in acutely dissociated Purkinje neurons (see Aman TK & Raman IM (2010) J Neurosci 30(16):5629).

3*) Abstract:*

*“These data suggest that the FGF14b N-terminus may serve as a blocking particle and/or cooperate with NaVbeta 4...” and Discussion as noted. There are no data on open channel block in this manuscript and this conclusion is unsupported. The data show nicely that reduction of functional FGF14 favors inactivation. Because any modulation that favors inactivation will reduce resurgent current, the observed reduction in peak resurgent current is predicted, though still worth documenting. The Results are presented quite clearly, but in the second paragraph of the Discussion the authors surprisingly and unnecessarily extrapolate to the suggestion that FGF14 is itself a blocker or forms part of a blocking complex*. *The existing data have no direct relationship to this conclusion, and actually seem to argue against the need to invoke this idea for the following reasons*:

*a) Losing FGF14 function speeds current decay at depolarized potentials, rather than slowing current decay at depolarized potentials, which is what would be expected if open channel block is directly disrupted (Khaliq et al. 2003, J Neurosci note also that these were the effects of trypsin and chymotrypsin in Grieco et al. 2005, Neuron, when the blocker was removed by proteolysis, although the authors argue that FGF14 might be a blocker because of trypsin sites)*.

*b) Loss of FGF14 reduces resurgent current but doesn't seem to change its kinetics, which might be expected if the interaction of the blocking particle with the channels is itself altered*.

*If the authors want to investigate whether FGF14 can block Na channels, they should simply test whether the N-terminus can make a resurgent current in channels that don't have a native blocker. In the absence of some kind of direct evidence of block, the arguments based on whether the protein has lysines or trypsin sites, etc. are unconvincing and irrelevant. Most importantly, the discussion of FGF14 as a blocker is incidental to the line of the paper and could be cut with no loss to the manuscript*.

As noted above, references to FGF14 as a blocking particle have been removed, as suggested.

4) Abstract:

*a) “Cerebellar Purkinje neurons fire repetitive action potentials because of a voltage-gated Na+ (NaV) ”resurgent“ current” That is maybe overstating the role of resurgent sodium current. There are neurons withour resurgent current that can still fire repetitively. Something like “Rapid firing of cerebellar Purkinje neurons is facilitated by voltage-gated Na+ (NaV) “resurgent” current...” would be more accurate*.

Changed as suggested.

*b) “Resurgent current results from unbinding of a blocking particle that previously terminated Na+ influx through the channel and simultaneously prevented channel inactivation.” This might be clearer as “Resurgent current results from unbinding of a blocking particle that competes with normal channel inactivation*.*”*

Changed as suggested.

5) Introduction:

*a) "Cerebellar Purkinje neurons, which provide the sole cerebellar output signal...“. Not true. A correct statement would be ”Cerebellar Purkinje neurons, which provide the sole output from the cerebellar cortex....” (not the cerebellum as a whole)*.

Changed as suggested.

*b) “This is an unusual behavior because, after initially opening, NaV channels in most neurons rapidly enter an inactivated state. This diminishes their ability to reopen quickly that defines the repetitive firing seen in Purkinje neurons.” This was a little clumsy. Something like “Fast repetitive firing in Purkinje neurons is promoted by unusual characteristics associated with sodium channel inactivation, whereby channels recover unusually rapidly from inactivation and in doing so pass a 'resurgent' sodium current, which helps drive the cell to fire a subsequent action potential*.*”*

Changed as suggested.

*c)“First, more than half of the NaV current in Purkinje neurons is carried by NaV1.6, an isoform that is particularly reluctant to enter the inactivated state (*[23]*).” This is a little misleading (because inactivation is still 95-99% complete) . Something like “..an isoform for which inactivation is somewhat less complete compared to others” would be better*.

Changed as suggested.

*6) “Although this result appears to contrast with a report that resurgent NaV current was unaffected by the absence of FGF14 in cerebellar granule neurons (*[5]*), the protocol employed in that study was markedly different. Instead of measuring resurgent current, it evaluated channel recovery from inactivation*.”

*This is incorrect.*
Table 1
*from Neuron 55(3):449-463 shows statistics for voltage clamp measurements of resurgent sodium current. However, as the measurements were in cerebellar granule cells, there is no real conflict. Also, the measurements reported were for resurgent current at -85 mV, which in the present results*
Figure 2
*showed much less change than at -40 mV where resurgent current is maximal*.

The reviewers are correct that Table 1 from Neuron 55(3):449-463 reports no change in “resurgent current”. From our reading of that publication, however, the data for Table 1 appear to be derived from the protocol and example traces described and depicted in Figure 7D. We believe that this protocol, and subsequently the data presented in Table 1 are therefore not representative of the protocols previously used to measure resurgent current.

*7) “FGF14 is expressed in the hippocampus (*[1]*), but resurgent NaV current is not observed in hippocampal neurons unless an exogenous blocking particle is added through the patch pipette (*[8]*)*.*”*

*This is true for CA3 pyramidal neurons but not CA1 pyramidal neurons, which do express resurgent sodium current (Royeck M, Horstmann MT, Remy S, Reitze M, Yaari Y, Beck H. Role of axonal NaV1.6 sodium channels in action potential initiation of CA1 pyramidal neurons. J Neurophysiol. 2008 100:2361-80*.*)*

We removed the discussion of FGF14 as a blocking particle, as suggested.

*8)*
Table 1
*includes several non-standard or ambiguous measurement parameters that are not defined in Methods, namely “AP rise time”, “AP decay time”, and “AP integral”. Because these do not change and are not referred to, they should be deleted or Methods should define how they were measured. (“AP rise time” is especially ambiguous and hard to interpret, because it is longer than AP duration*.*)*

Deleted as suggested.

*9) The authors write as though Purkinje neurons express only Nav1.6 channels. In fact, previous work suggests the presence of other sodium channels as well. For example, specific immunostaining and gene knockout indicate that Nav1.1 channels are present and constitute a significant fraction of sodium current in the cell bodies of dissociated Purkinje neurons (Kalume et al., J Neurosci, 2007). The presence of multiple sodium channel types in Purkinje neurons complicates all of the quantitative interpretations of the results presented here because it is unknown whether FHF-14 affects all of the sodium channels types that may be present in the same way*.

We agree that the interpretation of our data must account for all Na_V_ channels in Purkinje neurons and have revised both the Introduction and the Discussion accordingly. FGF14 also regulates Na_V_1.1, as previously shown [J Physiol 2005;569:179]. Thus, the effects on resurgent current that we attribute to FGF14 may result from FGF14’s overall influence of Na_V_ channel kinetics within Purkinje neurons. In addition to the added text, we removed the speculation that FGF14 may serve as a blocking particle for Na_V_1.6 (as suggested by the Reviewer); and with the consequent removal of the Na_V_1.6-centric model that was previously Figure 6, we believe that the revised manuscript addresses this issue appropriately.

*10)*
Figure 4*. Why are interactions of Nav1.5 and FHF-13 reported here? These results should be replaced with experiments on Nav1.6 and FHF-14, which form the primary focus of this manuscript*.

We designed our mutagenesis strategy to disrupt FHF-Na_V_ interaction by employing available crystallographic structural information from a complex that included FGF13 and the Na_V_1.5 C terminus. Because of the high sequence conservation among the FHFs and separately among the Na_V_1.x C termini, we believe that the FGF13-Na_V_1.5 model is relevant for Na_V_1.6 and FGF14. We agree with the Reviewer, however, that a demonstration that the FGF14b^R/A^ mutant cannot bind Na_V_1.6 will enhance our manuscript. Therefore, we expressed either FGF14b or mutant FGF14b^R/A^ with Na_V_1.6 in HEK cells and performed co-immunoprecipitation studies. As we now show in the new Figure 2, FGF14b^R/A^ does not co-immunoprecipitate with Na_V_1.6, demonstrating that the intact FGF14b^R/A^ does not interact with the intact Na_V_1.6.

*11) Figure 6. The molecular model developed in Figure 6 and the Discussion is highly speculative. There is no experimental basis for the interaction between the Navbeta4 subunit and FGF14 depicted in Figure 6, and indeed the role of Navbeta4 as a blocking protein has yet to be demonstrated for Purkinje cells. Similarly, the authors indicate that FHF-14 also is unable to generate resurgent sodium current when co-expressed with sodium channels, so there is no experimental basis for its proposed role as a blocking particle that generates resurgent sodium current either. If the authors wish to propose this molecular mechanism, they need to show that co-expression of Navbeta4, FHF-14, or both can generate resurgent sodium current in a cell type in which none is observed before co-expression*.

As suggested above, the references to FGF14 as a blocking particle have been removed.

*12) The authors base much of the impact of their manuscript on elucidating the molecular mechanism of resurgent sodium current. However, as developed in the points above, their work does not directly address this problem. The authors need to re-orient the Abstract and Discussion and provide a more persuasive presentation of the importance of their findings for the physiology of Purkinje neurons and the pathophysiological effects of spinocerebellar ataxia mutations in FHF-14. The discussion of the molecular role of FHF-14 and Navbeta4 should be greatly reduced in favor of consideration of the potential impact of regulation by FHF-14 in the physiological function and pathophysiology of Purkinje neurons*.

As suggested, the manuscript has been re-tooled (particularly in the Discussion) to discuss the role of FGF14 as regulator of resurgent current through its effects upon Na_V_ channel kinetic properties. All references to FGF14 as a potential blocking particle have been removed. The resultant discussion focuses more on the roles for FGF14 in Purkinje physiology and the consequences for spinocerebellar ataxia.